# The Epidemiology of Skin Cancer and Public Health Strategies for Its Prevention in Southern Africa

**DOI:** 10.3390/ijerph17031017

**Published:** 2020-02-06

**Authors:** Caradee Y. Wright, D. Jean du Preez, Danielle A. Millar, Mary Norval

**Affiliations:** 1Environment and Health Research Unit, South African Medical Research Council, Pretoria 0001, South Africa; Danielle.Millar@mrc.ac.za; 2Department of Geography, Geoinformatics and Meteorology, University of Pretoria, Pretoria 0002, South Africa; dupreez.dj@tuks.co.za; 3LACy, Laboratoire de l’Atmosphère et des Cyclones (UMR 8105 CNRS, Université de La Réunion, Météo-France), 97744 Saint-Denis de La Réunion, France; 4Biomedical Sciences, University of Edinburgh Medical School, Edinburgh EH8 9AG UK; mary.norval@ed.ac.uk

**Keywords:** climate change, environmental health, HIV/AIDS, keratinocyte cancer, melanoma, oculocutaneous albinism, public health, sun exposure

## Abstract

Skin cancer is a non-communicable disease that has been underexplored in Africa, including Southern Africa. Exposure to solar ultraviolet radiation (UVR) is an important, potentially modifiable risk factor for skin cancer. The countries which comprise Southern Africa are Botswana, Lesotho, Namibia, South Africa, and Swaziland. They differ in population size and composition and experience different levels of solar UVR. Here, the epidemiology and prevalence of skin cancer in Southern African countries are outlined. Information is provided on skin cancer prevention campaigns in these countries, and evidence sought to support recommendations for skin cancer prevention, especially for people with fair skin, or oculocutaneous albinism or HIV-AIDS who are at the greatest risk. Consideration is given to the possible impacts of climate change on skin cancer in Southern Africa and the need for adaptation and human behavioural change is emphasized.

## 1. Introduction

Sun exposure is the major environmental risk factor for skin cancer [1]. Cutaneous melanoma (CM) is fatal if untreated while the keratinocyte cancers (KCs), namely basal cell carcinoma (BCC) and squamous cell carcinomas (SCC), are disfiguring and debilitating [2]. Globally, CM is the 19th most common cancer worldwide [3] and KCs have the highest incidence of any cancer in the Caucasian population. In 2018 in the world, there were about 300,000 new cases of CM recorded, and more than 1 million new cases of KCs, although the latter is likely to be a gross underestimate due to challenges related to diagnosis and reporting. For example, in the United States a total of 5.4 million KCs were diagnosed in 2012 [4].

Generally, little has been published about skin cancer prevalence and epidemiology in Southern Africa. Here the geography of this part of Africa is first described, followed by an overview of the solar ultraviolet radiation (UVR) environment across the region. Then, for each country, the epidemiology of CM and KC are outlined. Risk factors for the development of skin cancer including exposure to solar UVR, skin phototype, the prevalence of HIV/AIDS and oculocutaneous albinism (OCA) are considered. An attempt is then made to explore the existence and nature of public health campaigns for skin cancer prevention in Southern African countries. These findings are used to suggest recommendations for reducing the risk of skin cancer and encouraging sun protection for individuals at greatest risk. Finally, the impacts of climate change on skin cancer in Southern Africa are considered.

## 2. Geography and Population

Southern Africa consists of five countries: Botswana, Lesotho, Namibia, South Africa, and Swaziland (recently renamed as eSwatini) (Figure 1) [5]. Coastal low-lying areas develop inland into mountain ranges that lead to a plateau at high altitude.

The population characteristics of the five countries are outlined in Table 1. South Africa has the largest population of nearly 60 million while Botswana has the highest life expectance of 67 years. All the countries except South Africa have less than 5% of the population who are not Black African.

## 3. Meteorology and Solar UVR Environment

The climate regions of Southern Africa are seasonal and include arid to subtropical areas. Southern Africa is affected by extreme weather events [8]. The altitude, climate, and solar UV Index (UVI: a metric to describe the level of solar UVR at the Earth’s surface with values above 11 considered extreme) for the capital cities of the five countries are shown in Table 2.

Solar UVR levels across Southern Africa depend on factors such as latitude, altitude, ozone, aerosols (e.g., sea salt spray, air pollution), albedo (reflection from snow or water) and cloud cover. Summer (December, January, February) averages for 2009–2018 for solar noon UVI from the satellite-based ozone monitoring instrument (OMI) [10] show that the entire region experiences values of 10 or higher (Figure 2). In the winter months, however, the highest latitude regions experience moderate levels and countries nearer to the Equator experience higher levels with a maximum of 9 UVI. Over Southern Africa, total column ozone is at a maximum during the austral spring and decreases over the summer months when UVR is at a maximum [11]. More southerly areas experience a higher variability of ozone throughout the year [12].

## 4. Introduction to Keratinocyte Cancer (KC) and Cutaneous Melanoma (CM)

In general, KCs are rare in children, and BCCs are more frequent in middle age than SCCs, with approximately equal numbers in old age [13,14]. The incidence of both BCC and SCC has been steadily increasing globally in countries with predominantly fair skinned populations through much of the past sixty years or so [15]. This is most likely due to changes in the prevalence of risk factors, as outlined below in Section 7, in conjunction with more awareness and early detection. Although the KCs are rarely fatal, they can be disfiguring and lead to considerable medical costs relating to histological diagnosis and treatment.

CM is found at a lower incidence than the KCs in those with fair skin. The incidence rate increases with age. Temporal trends in incidence are variable with data from some countries showing a steady increase over the past hundred years, such as in the United Kingdom [16], while, in other countries, such as Australia, following a considerable rise, a decrease is indicated in younger age groups in recent years [17]. Unlike most KCs, CM, if undiagnosed at an early stage, can prove fatal. However, age-standardised mortality rates due to CM have stabilised in several countries. For example, the mortality rate in Australia remained unchanged at 6.0 per 100,000 between 2004 and 2012 [17]. This improvement is likely to be due to earlier detection and new treatments involving immunotherapy and immune checkpoint inhibitors.

## 5. The Incidence of Skin Cancer in Southern Africa

The five countries which comprise Southern Africa do not have robust National Cancer Registries for assessing the incidence of KCs and CM (see Table 3). Lesotho has no Registry at present and cases in Swaziland are not necessarily reported centrally. The Botswana National Cancer Registry is population-based: it collects information through cancer notifications and discharge forms, an electronic integrated patient management system, and visits by registry personnel to referral hospitals. The registry in South Africa reports only on pathology-confirmed cases while Namibia’s is pathology-based and also includes cases diagnosed clinically. Therefore, the figures for new cases of KCs and CMs arising in any one year in these countries cannot be considered accurate, with under-reporting to be very likely.

The most recent numbers of KC and CM cases available for each of these countries are shown in Table 3. Small numbers are seen for Botswana and Swaziland, with higher numbers for Namibia, perhaps due to the more comprehensive method for collecting data in this country. In South Africa the numbers are much higher, a reflection of the relative population size (Table 1), the higher percentage of people with fair skin in the population compared with the other four countries (Table 1), and possibly the provision of more extensive medical care and personal awareness. Figures giving the age-standardised incidence rate of KC per 100,000 in Southern Africa (16.4 for men, 7.2 for women) were estimated in 2018 [19]. These are higher than reported for the other regions in Africa which are generally around two for both men and women [19] and may represent the larger percentage of people with fair skin in the population of South Africa compared with most of the other countries in the African continent. In support, data obtained between 2000 and 2004 in South Africa found that the mean age-standardised incidence of BCC was 3.0 and 1.7 in Black African men and women respectively, while it was 198 and 113 in White men and women respectively; for SCC, it was 3.0 and 1.6 in Black African men and women respectively, while it was 70 and 32 in White men and women respectively [20]. As figures for the number of new cases of KC per year are not considered accurate, any trend in incidence cannot be established.

Regarding the incidence of CM, data are available only for South Africa out of the Southern African countries. The age standardised (world standard population) incidence rate per 100,000 was 4.9 in men and 2.9 in women in 2013. In the White population only, it was 19.7 in men and 13.8 in women [21].

## 6. Mortality due to Skin Cancer in Southern Africa

Deaths due to skin cancer in the countries comprising Southern Africa are estimates at best due to the lack of recording accurate data in past years. Figures have been compiled, largely based on incidence, which give the number of skin cancer deaths in 2017, together with the age-adjusted skin cancer deaths per 100,000 in each country (Table 4). A distinction between death due to KC or CM was not made although it is likely that the majority was due to CM with a minority due to SCC, and none to BCC. In 2016, Statistics South Africa reported 456,612 deaths; 826 from CM and 485 from ‘other disorders of the skin and subcutaneous tissue’ which includes KCs. Thus, about twice (58%) as many skin cancer deaths were reportedly from CM than KCs that year in South Africa [22]. It should also be remembered that the provision of medical care is unlikely to be the same in each Southern African country.

CM deaths in South Africa in each year between 1997 and 2014 were assessed recently to try to establish if there were any trends [23]. It was found that there was an increasing trend in age-adjusted mortality rates in men between 2000 and 2005, rising from 2 to 3 per 100,000, and in women between 1997 and 2014 from 0.9 to 1.2 per 100,000. An increase in the White population, rising by 3% over the period 1999 to 2014, was also apparent, but no trend was seen in the Black African population group. It was acknowledged that the lack of a comprehensive population-based death registration limited the validity of the results obtained, as well as possible under-reporting and uncertainty about assigning population group categories. Despite these reservations, this study was the first to attempt an analysis of mortality trends due to CM in any African country.

## 7. Risk Factors for the Development of Skin Cancer in Southern Africa

In Black Africans, SCCs are the most frequent dermatological malignancy [24] and are found commonly on the lower leg in sites of chronic ulceration and previous burns, which are not sunburns. CMs are the next most common in Black Africans [25] and are mainly acral (on palms of hands, soles of feet and around nails), with the majority located on the foot. They tend to be diagnosed late and are more aggressive than CMs in those with fair skin [25,26]. BCCs are the least frequent, but occur on similar body sites, that is on sun-exposed areas and on the trunk, as in those with fair skin. CMs and SCCs have a different distribution in fair skinned populations compared with Black Africans [21] as CMs are considerably less common than the KCs, SCCs occur mainly on body sites most exposed to the sun such as the face and backs of the hands, while the body sites of CMs are variable, occurring most frequently on the back in men, the legs in women, and on sun-exposed body sites in older people. Such variation implies that risk factors for skin cancer development may differ between population groups with varying skin colour.

The major environmental risk factor in people with fair skin is exposure to the sun [27]: cumulative life-time exposure for SCC, and intermittent intense exposure, particularly in childhood and adolescence, for BCC. This is emphasised by the high association between occupational solar UVR exposure in outdoor workers and the incidence of KC [28]. For CM, a dual pathway is likely, that is, one involving naevi initiated by early sun exposure and promoted by intermittent sun exposure, and the other involving chronic sun exposure in sun-sensitive individuals [29]. In people with pigmented and deeply pigmented skin, the high content of cutaneous eumelanin which gives the skin its brown-black colour, provides protection from the sun against the development of skin cancer, estimated as a 13.4-fold decrease in African Americans [30]. Thus, while sun exposure is very likely to be a risk factor for SCCs and BCCs in Black Africans as in the White population, incidence figures from South Africa, outlined in Section 5 above, demonstrate the considerable protection offered by eumelanin in Black Africans.

As CMs in Black Africans are mainly acral and thus occur on nonexposed body sites, risk factors other than sun exposure are likely to be more important. It is possible, for example, that they develop on pre-existing melanocytic naevi found on acral surfaces without a direct role for solar irradiation. Alternatively, an indirect role may be possible as sun exposure is known to cause systemic immunosuppression by the release of a range of mediators from an irradiated site [31]. Such mediators could act at sites distant from the irradiated site, causing significant downregulation in cell-mediated immune responses [31]. Such immunomodulation also occurs in those with fair skin and indeed follows exposure to lower UVR doses than in those with pigmented skin [32]. Whether there are differences in the type and quantity of these immune mediators following irradiation of skin of different colours is not known at present.

A second risk factor for skin cancer development is OCA which occurs in a proportion of the Black African population in Southern Africa. The prevalence was last recorded in Botswana in 1987, in South Africa in 2006, in Namibia in 2011 and in Lesotho in 2014 (Table 1). More recent figures have not been published. In OCA, which is an inherited autosomal recessive condition, synthesis of melanin is considerably reduced leading, in most instances, to creamy White skin and light-coloured eyes, typically light ash to brown. Sunburn is reported frequently in children with OCA resulting in chronic skin damage and multiple solar keratoses (precursor lesions to SCC) [33]. Individuals with OCA have a thousand times higher risk of developing skin cancers, mainly SCCs, compared with the general population by the age of 20–30 years [34]. Such tumours progress rapidly, metastasise, and can result in a lower life expectancy. The role of solar irradiation in this process is emphasised as a higher frequency of SCCs is found in OCA individuals living nearer the equator compared with those living at higher latitudes in South Africa [35].

A third risk factor is the prevalence of HIV/AIDS in Southern Africa which is amongst the highest in the world at approximately 25% of the population in Botswana, Lesotho, and Swaziland, with lower percentages in South Africa and Namibia (Table 1). Infection with HIV leads to impaired immunity which might, in turn, result in less effective control of the initiation or progression of skin tumours. Indeed, it is known that CMs in HIV-positive patients are more aggressive and survival rates are poorer than in HIV-negative people [36]. Studies in California demonstrated that there was a greater than two-fold higher incidence of SCC, but not of BCC, in White HIV-positive patients compared with White HIV-negative controls [37,38]. An equivalent study in Denmark revealed a five-fold increase in SCC rate and a 1.8-fold increase in BCC in HIV-positive patients [39]. Recent reports investigating the risk of CM in White HIV-positive patients in North America [40] and Denmark [39] found no difference compared with the general population. As is the case for CM in immunocompetent individuals, older age and higher sun exposure were associated with an increased risk [40]. It is of interest that CM incidence increased slightly in those patients treated with antiretroviral therapy for the previous two years [40]. This was suggested to be due to regular skin surveillance and not to the medication. Equivalent studies have not been done in Southern Africa although Jaquet et al. [41] reported that HIV infection was associated with a five-fold increase in SCC incidence in four West African countries. Therefore, it is likely that the high prevalence of HIV in Southern Africa may lead to an increased risk of SCC in particular.

Another common skin tumour in HIV-positive individuals is Kaposi’s sarcoma, although its incidence has declined markedly since the advent of antiretroviral therapy. There is limited evidence from epidemiological studies to indicate that exposure to solar UVR increases the risk of Kaposi’s sarcoma [42].

Age requires consideration as a risk factor as skin cancer develops predominantly in older age. For example, BCC occurred at a mean age of mid-60s in both men and women in South Africa, and CM at a mean age of mid-50s in the Black African, Coloured, Asian and White population groups [20]. Table 1 shows the average life expectancy in each of the countries comprising Southern Africa. All are at the lower end of the ranking of life expectancy in the countries of the world [43], partly due to common infections, particularly HIV and tuberculosis, to poor economic and social conditions, and to limited medical resources. The impact of lower life expectancy on the prevalence of skin cancers and age of diagnosis is largely unknown. One attempt has been made to address this question as follows. As the average age at diagnosis of SCC in Black Africans in South Africa is about ten years younger than in the White and Coloured population groups, Diffey et al. [44] demonstrated that the prevalence of HIV infection in the Black African population decreased the age at SCC diagnosis compared with the White and Coloured population groups.

There is a small contribution to an increased risk of developing CM by having particular genetic polymorphisms, and it is known that a family history of CM increases risk by an estimated 74% [45]. Systematic genetic analyses to tests for such traits have not been performed on Southern African populations.

A final risk factor suggested for SCCs in Black Africans, particularly those living in rural or poor communities, may be trauma as many of these tumours arise in such sites on the legs [46]. There is little evidence to support this suggestion.

## 8. Campaigns to Prevent Skin Cancer in Southern Africa

Public health campaigns to raise awareness about excess sun exposure and to protect against skin cancer have existed all around the world for several decades. Australia’s SunSmart programme is one of the longest-standing, multi-component community-wide skin cancer prevention programmes in the world [47]. Recently, the effectiveness of this campaign was assessed and showed that, with an average 20-year lag between sun exposure and CM development, the SunSmart campaign contributed to a reduction in CM among younger individuals living in Melbourne [47]. Moreover, a study in Belgium found that sun safety campaigns can be cost-effective in reducing the incidence of skin cancer [48].

Notwithstanding the value of skin cancer prevention campaigns, there is a potential conflict regarding safe sun exposure to ensure UVR-induced production of vitamin D, a hormone essential for bone health [49]. Vitamin D insufficiency has been associated with autoimmune diseases and some cancers, as well as mental health disorders [50]. Factors that limit UVR-induced vitamin D production include covering the body with clothing and an indoor lifestyle. Thus, a balanced approach to sun exposure is important for holistic health and wellbeing. A review published in 2016 indicated that vitamin D status among different South African population groups and ethnicities was generally sufficient in children and adults, but was insufficient in many individuals over the age of 65 years [51].

In Southern Africa, information about skin cancer and its prevention and local campaigns to raise awareness about skin cancer that are available on the Internet (and therefore retrievable) are most common in South Africa and Namibia (see Appendix A). The Cancer Association of South Africa has a dedicated website that explains the need for protection against excess sun exposure. Two other organisations also promote skin cancer prevention in South Africa. One is the Skin Cancer Foundation of South Africa which runs an annual free screening in Spring. The other, the Albinism Society for South Africa, advocates sun protection for people with OCA in the country. Namibia’s Cancer Association is available on the Internet, and a recent newsletter (published in August 2019) included an article on ‘How to Protect Your Skin from The Sun’. There is no Internet presence for a Cancer Association or information relating to specific skin cancer prevention available for Botswana (except for a page giving contact details for the Botswana Cancer Association), Lesotho or Swaziland. Access to the Internet in Southern Africa ranges between 47% and 56% of the population (see Table 1, last row). This low access and usage rate could be the reason why sun protection messages are not available online. Other communication media, such as radio and text messages that can be read on basic mobile phone, have been promoted by the World Health Organization as important communication media in Africa [52], and these may be useful alternatives.

In all Southern African countries, messages about skin cancer prevention together with screening and early detection should be made available via communication media that can reach as many people as possible. Schools are one setting in which most children can access health information and government health programmes should include sun protection and skin cancer prevention advice. Radio, and to some extent free-to-air television, are the main communication tools in Southern Africa, and not the Internet. While many people have a mobile phone, most of these are not smartphones with Internet access, which is expensive in Southern Africa (up to 10% of a monthly income) [53]. However, data costs are likely to decrease and smartphone usage increase in the coming years and so Internet-based, local messages about sun protection in Southern African countries may be important for the future.

Given the prevalence of OCA in Southern Africa (see Table 1), some countries promote sun protection to reduce excess solar UVR exposure specifically in people with OCA. The Cancer Association of South Africa has a dedicated webpage that gives information about why those with albinism need sun protection. There is a similar website in Lesotho (Appendix A) and an entity called ‘Support in Namibia of Albinism Sufferers Requiring Assistance’ assists Namibians with OCA to protect themselves from excess sun exposure.

In general, the online presence of cancer organisations that provide information about personal photoprotection, prevention strategies, palliative care and family support is inconsistent in Southern African countries. Such bodies could draw on the World Health Organization website [54] on skin cancer prevention, preferably provided in the local languages of the country.

## 9. Climate Change and Skin Cancer in Southern Africa

Southern Africa is considered one of the most vulnerable regions in the world to the effects of climate change. Current climate models show a positive trend in surface temperatures and temperatures are expected to increase by approximately 3–5 °C, although this increase is not temporally or spatially uniform across the region [55]. Countries like Namibia and the north-western parts of South Africa are predicted to experience a larger increase in temperatures.

Projections for Southern Africa suggest that the number of hot days (≥27 °C) will increase [56]. How warmer temperatures will affect skin cancer in Southern Africa is unknown. There is some evidence that skin cancer rates may increase with a rise in ambient temperature, although the reasons why are not fully understood [57]. Estimates made in 2018 suggest a modest 5.8% increase in SCC and a smaller 2.8% increase in BCC according to ambient temperature increases projected between 2018 and 2100 [57,58].

In warmer conditions, behavioural patterns may change and affect personal solar UVR exposure. People may wear less clothing or partake in activities such as swimming to try to cool down. They may spend more time outdoors, especially if indoors is warmer than outdoors, as has occurred in dwellings in urban Johannesburg [59]. Similarly, in schools where shipping containers are being used as classrooms, indoor temperatures can exceed 40 °C [60]. In such hot conditions, schoolchildren experience tiredness and breathing difficulties and teachers may prefer to take children outdoors [60]. This may lead to increased exposure to the sun and possibly increase skin cancer risk. Anecdotal evidence and observations suggest shipping containers are being used more often than in the past. No information about how many schools have air conditioners is available; moreover, with the current electricity crisis in the country (i.e., load-shedding or rolling black-outs), it is likely that the few schools with air conditioning would not be able to use them regularly.

In hot conditions, as was found in Austria when temperatures exceeded 30 °C [61], people may spend more time indoors and or seek shade when outdoors. While it is unknown how those living in Southern Africa will respond to hotter conditions, evidence suggests that the time-lapse between changes in ambient temperature and skin cancer incidence is about 60 years [62,63]. Hence, the full extent of the impact will not become apparent for some time.

## 10. Sun Protection Strategies

Solar UVR exposure is the only modifiable risk factor for skin cancer. The World Health Organization recommends five steps to protect against excess sun exposure [64]. First, to limit the time spent in the midday sun between 10:00 and 16:00. Secondly, to note the UVI as an aid in planning outdoor activities, although not to rely solely on the information [65]. Thirdly, individuals are advised to use shade wisely and to remember the shadow rule: “Watch your shadow: short shadow, seek shade!”. The third and fourth messages are to wear protective clothing, such as wide-brimmed hats, sunglasses and tightly woven, loose-fitting clothes, and to use sunscreen. The sunscreen should be broad-spectrum, have a sun protection factor of at least 15+ and be re-applied every two hours, or after swimming, working, playing or exercising outdoors. The fifth step is to avoid using sunlamps and tanning devices. These sun protection messages need to be conveyed particularly to vulnerable individuals, such as people with OCA or fair skin, or in high risk settings. They should be presented in a way that is easy to understand and in the local language of the country, as discussed below.

Schools can be high sun exposure environments [66]. Outdoor activities such as break times, and sporting and cultural events are often scheduled during peak UVR hours. Instead, they should avoid the sun at these intense times or at least be situated in locations with adequate shade cover. The planning and implementation of all aspects of sun protection in schools are required in Southern Africa since evidence from elsewhere suggests that they are effective [67]. This means curriculum content on sun protection, role modelling of sun protection by school staff, provision of accessible shade, advocating sun protective clothing and hats and use of sunscreen. The cost of sun protection items, especially sunscreen, is often a barrier to their use and who bears these costs needs consideration [68].

Sun protection is critical for people with OCA and other individuals with fair skin. In South Africa, sunscreen with SPF 15 is available from government clinics and hospitals free of charge for those with OCA. Despite this, children with OCA do not make use of the government-supplied sunscreen because trips to the primary healthcare facilities are time-consuming with inadequate availability and insufficient supply [69]. An alternative may be to supply sunscreen to schools where children with OCA are enrolled.

Outdoor workers in, for example, agriculture, construction, mining and roadworks, experience high levels of sun exposure. Personal protective equipment (PPE) issued to outdoor workers by their employers is typically long-sleeved shirts and trousers. The fabric of the material needs to be thermally comfortable while providing sun protection. Hats or helmets with flaps and sunglasses should also be part of the occupational PPE. Patterns of sun exposure in outdoors workers have been assessed in limited studies in South Africa [70,71] while numerous studies around the world have considered patterns of sun exposure and risk of adverse health effects [72,73,74].

High risk settings in which people with fair skin who are at relatively high risk of skin cancer include recreational locations such as the beach and swimming pools, parks, gardens, and sports grounds. Risk communication messages using tools, like the UVI, are useful here. For example, the Pool Cool Program in the United States [75] used staff training, sun safety lessons, sunscreen, shade and signage to create safe pool environments. Positive changes in both parents’ and children’s sun protection habits and reduced incidence of sunburn occurred. Similar programmes may be beneficial in Southern Africa.

## 11. Conclusions

Personal solar UVR exposure depends on geographical and behavioural factors while subsequent adverse health risks from excess exposure are influenced by individual susceptibility, such as skin phototype and genetic disposition. In Southern Africa, the incidence of skin cancer differs between the countries in this region by population group and gender. Studies in several other countries have shown an increase in skin cancer incidence in the past fifty years or so in individuals with fair skin and increasing age. Equivalent statistics for the population groups in Southern Africa are lacking. While it is known that the content of eumelanin in Southern Africans with deeply pigmented skin exerts a protective effect against the risk of skin cancer, this population group is not precluded from developing such cancer. In individuals with OCA, the risk of skin cancers is greatly enhanced, although more data are required to inform sun protection policy and practice for this group. The prevalence of HIV/AIDS also increases susceptibility to skin cancer depending on sun exposure patterns and in an age-dependent manner.

While limited information on the epidemiology of skin cancer in Southern Africa exists, collection of reliable and comprehensive national statistics regarding incidence and mortality would give more insight for public health planning. There are differences in the method of reporting skin cancer incidence between the Southern African countries, thereby making the data too uncertain to establish any trends. Underestimation of the rate of skin cancer incidence is also likely. From available online evidence, it seems that limited educational campaigns and resources are available for sun protection in Southern Africa. In a changing climate with an increase in skin cancer incidence possible, the need for sun protection programmes in local languages, using accessible media such as radio, as well as affordable protective clothing and shade will be important.

## Figures and Tables

**Figure 1 ijerph-17-01017-f001:**
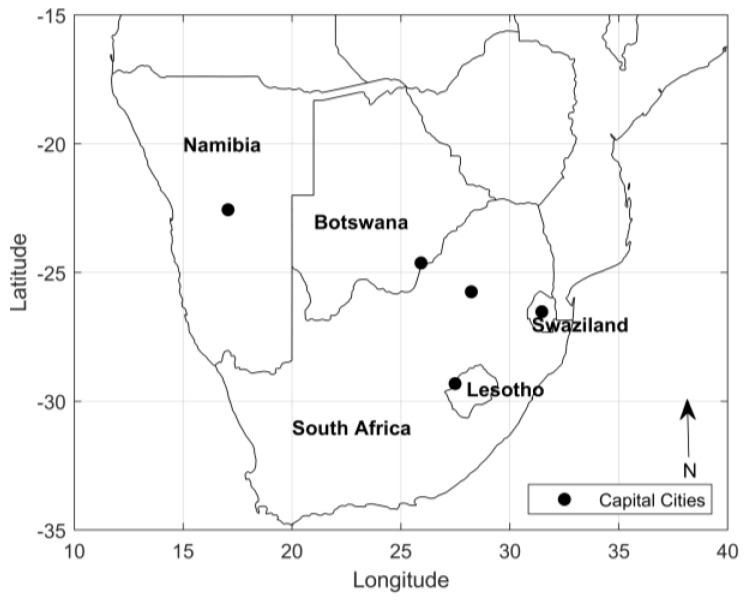
Map of Southern Africa.

**Figure 2 ijerph-17-01017-f002:**
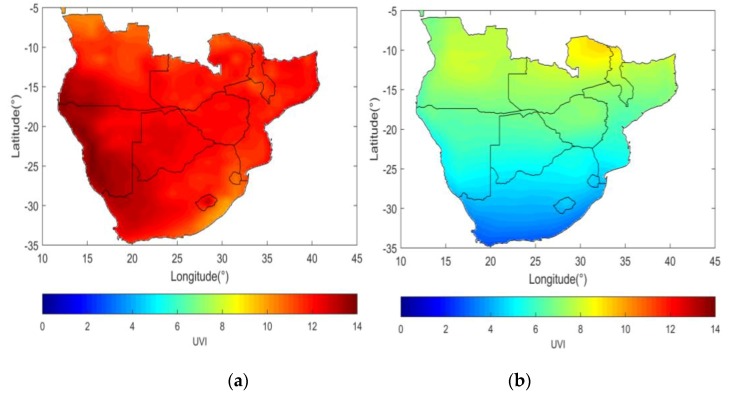
Solar-noon UV Index values for (**a**) summer and (**b**) winter in Southern Africa.

**Table 1 ijerph-17-01017-t001:** Population statistics for the countries comprising Southern Africa [6,7].

Indicator	Botswana	Lesotho	Namibia	South Africa	Swaziland (eSwatini)
Number in population (2018)	2.3 million	2.1 million	2.5 million	58.8 million	1.4 million
Percent population not Black African	1.0%	0.3%	4.0%	9.2% White; 8.8% Coloured; 2.6% Asian/Indian	3.0%
Life expectancy, years (2018)	67	54	64	63	58
Prevalence HIV/AIDS in adults (2016–2018)	25.0%	24.0%	12.6%	19.0%	27.2%
Death rate due to HIV/AIDS per 100,000 (2017)	203.3	648.0	170.1	273.1	430.3
Prevalence of albinism	1 in 2243 in Southern Sotho people (in 1987)	1 in 6000 (in 2014)	1 in 2168 (in 2011)	1 in 3900 (in 2006)	Not known
Access to and usage of Internet	47%	30%	51%	56%	47%

**Table 2 ijerph-17-01017-t002:** Meteorological characteristics of five largest cities in Southern Africa including their average noon UV Index [9].

	Botswana	Lesotho	Namibia	South Africa	Swaziland (eSwatini)
Capital city	Gaborone	Maseru	Windhoek	Pretoria	Mbabane
Estimated population in capital city (millions)	0.2	0.1	0.3	1.6	0.1
Latitude	24.6° S	29.3° S	22.6° S	25.7° S	26.5° S
Altitude (metres)	1010	1600	1655	1450	1200
Summer mean noon UV Index	12	13	13	12	10
Winter mean noon UV Index	5	4	7	5	3
Mean day-time temperature in summer (°C)	33	27	32	29	25
Mean day-time temperature in winter (°C)	23	15	22	20	20

**Table 3 ijerph-17-01017-t003:** Number of cases of keratinocyte cancer (KC) and cutaneous melanoma (CM) in the countries of Southern Africa [18]. No figures available from Lesotho where the first National Cancer Registry is due to open in 2024.

	Botswana (in 2009–2013)	Namibia (in 2013–2015)	South Africa (in 2010–2014)	Swaziland (eSwatini) (in 2016–2017)
Number of cases (KC)				
male	176	391	59975	35
female	153	285	42550	44
Number of cases (CM)				
male	42	65	3583	5
female	71	93	3243	9
Year National Cancer	1999	1995	1986	2015
Registry established	Population based	Pathology based plus clinical cases	Pathology based	Cases not reported centrally

**Table 4 ijerph-17-01017-t004:** Deaths due to skin cancer in the countries of Southern Africa in 2017 [6].

	Botswana	Lesotho	Namibia	South Africa	Swaziland (eSwatini)
Number of skin cancer deaths	46	47	19	1659	25
Age-adjusted skin cancer deaths per 100,000	4.0	4.2	1.6	4.5	4.3

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
