# Peer review of "The Epidemiology of Skin Cancer and Public Health Strategies for Its Prevention in Southern Africa"

_ijerph, 2020, doi:10.3390/ijerph17031017_

Round 1

Reviewer 1 Report

This paper provides what appears to be the first overview of the epi of skin cancer and prevention strategies (in the context of climate change) in Southern Africa. Specific suggestions for manuscript improvements are provided below. These primarily include recommendations for clarifications and additional detail in places.

Title: Suggest changing “measures” to “strategies” or something similar since the paper is not focused on assessment measures, which is another common meaning.

Intro: “In 2018 in the world, there were about 300 000 new cases of CM recorded, and about 1 million new cases of KCs, although the latter is likely to be an underestimate due to challenges related to diagnosis and reporting [4]”. The KCs seem to be a gross underestimate since almost 5 million new cases in the USA have been estimated annually in recent years. Please also review the references, because the sentence refers to 2018, but the article cited was published in 2017.

Meteorology: Table 2 – This table is confusing because the title refers to the 10 largest cities, but only five capital cities are mentioned by name. Please clarify in the table or text. Based on Figure 2, it seems that the highest latitude regions experience moderate rather than low UVI.

Intro

Suggest briefly listing the risk factors in this section that are discussed in more detail in section 7.

Incidence

Please provide more information about the different types of case reporting, particularly population-based, which was not described.

Risk factors Suggest specifying whether the previous burns were likely sunburns or not. Please expand and add a few citations related to late diagnosis and aggressiveness in darker-skinned populations. “Such variation implies that risk factors for skin cancer development may differ between population groups with varying skin colour.” Suggest adding “and/or sex” to the end of the sentence. “… provides protection against the development of skin cancer”. Please add “from the sun” to provides protection. It is suggested that immunosuppression may play a role in Black Africans. Would it not be similar in Whites? Please clarify and provide citations for any likely differences. Suggest specifying typical colors of “light-coloured eyes” among individuals with OCA.

Campaigns What is known about Vitamin D deficiencies in Southern Africa or among Blacks elsewhere? When it is said that campaigns are “available on the Internet”, does that mean referred to or cited on the Internet, delivered to the public via the Internet, or both? Perhaps text message interventions could be suggested as an option since it seems that more people have cell phones than Internet access, and texting interventions have shown promise elsewhere.

Climate “Projections for Southern Africa suggest that the number of hot days (≥27°C) will range between 81 and 296 per year, relative to 1961-1990, by 2100 [45].” This sentence is unclear unless it is meant that the number of hot days will increase, rather than range, since no comparison is provided. “The estimates range from a modest 5.5% increase in SCC and a 2.9% increase in BCC per °C increase [46] to a 10% increase for every 2°C increase in ambient temperature [47].” Please revise this sentence to clarify it. I was initially confused because 5.5x2~10, but then I realized the 10% was probably referring to BCC only. What is the prevalence of the use of shipping containers and/or air conditioning in schools? “…evidence suggests that the time-lapse between changes in ambient temperature and skin cancer incidence is about 60 years [51].” The paper cited here is quite old.

Sun protection Some recommendations specify 30+ for sunscreen SPF. What is the prevalence of sunlamp use in Southern Africa? “Sun protection is critical for people with OCA.” Suggest adding “and other individuals with fair skin”. Are there any data on outdoor workers in Southern Africa? Caution about the use of the UVI as a skin cancer prevention tool for the public is warranted. See Prev Med.2019 Jun;123:71-83. doi: 10.1016/j.ypmed.2019.03.004. Epub 2019 Mar 4. Awareness, understanding, use, and impact of the UV index: A systematic review of over two decades of international research. Heckman CJ1, Liang K2, Riley M3. It seems that HIV+ individuals would be a potential target for sun protection efforts. Is it worth mentioning other skin cancers common among them such as kaposi’s sarcoma?

Author Response

IJERPH-684722 Response to Reviewers: The epidemiology of skin cancer and public health measures for its prevention in Southern Africa

Reviewer 1

This paper provides what appears to be the first overview of the epi of skin cancer and prevention strategies (in the context of climate change) in Southern Africa. Specific suggestions for manuscript improvements are provided below. These primarily include recommendations for clarifications and additional detail in places.

Thank you for taking the time to review our manuscript and to provide useful comments and suggestions.

Title: Suggest changing “measures” to “strategies” or something similar since the paper is not focused on assessment measures, which is another common meaning.

We have considered your suggestion and amended the title to “strategies”.

Intro: “In 2018 in the world, there were about 300 000 new cases of CM recorded, and about 1 million new cases of KCs, although the latter is likely to be an underestimate due to challenges related to diagnosis and reporting [4]”. The KCs seem to be a gross underestimate since almost 5 million new cases in the USA have been estimated annually in recent years. Please also review the references, because the sentence refers to 2018, but the article cited was published in 2017.

We have amended the estimate of KCs and changed the reference to the 2012 US estimate (published in 2015) as suggested to emphasize the gross underestimation.

Meteorology: Table 2 – This table is confusing because the title refers to the 10 largest cities, but only five capital cities are mentioned by name. Please clarify in the table or text. Based on Figure 2, it seems that the highest latitude regions experience moderate rather than low UVI.

We have amended the number of cities to five and corrected the text in line 14 that moderate levels are experienced.

Intro

Suggest briefly listing the risk factors in this section that are discussed in more detail in section 7.

The major risk factors are already listed in the middle of the second paragraph.

Incidence

Please provide more information about the different types of case reporting, particularly population-based, which was not described.

Information has been added to the text to explain what the population-based cancer registry in Botswana consists of.

Risk factors

Suggest specifying whether the previous burns were likely sunburns or not.

A sentence has been added indicating that the burns were not due to sunburn.

Please expand and add a few citations related to late diagnosis and aggressiveness in darker-skinned populations.

Two references have been added which give further details.

  “Such variation implies that risk factors for skin cancer development may differ between population groups with varying skin colour.” Suggest adding “and/or sex” to the end of the sentence.

Data are not available to assess gender differences in the incidence/body site of skin cancers in the countries of Southern Africa beyond societal differences such as in clothing or occupations. Therefore the suggestion to add “and/or sex’ has not been done.

 “… provides protection against the development of skin cancer”. Please add “from the sun” to provides protection.

“from the sun” has been inserted, as suggested

It is suggested that immunosuppression may play a role in Black Africans. Would it not be similar in Whites? Please clarify and provide citations for any likely differences.

More information is now inserted at the end of the third paragraph in section 7, together with a new reference.

 Suggest specifying typical colors of “light-coloured eyes” among individuals with OCA

The eye colours are now added to the text.

Campaigns

What is known about Vitamin D deficiencies in Southern Africa or among Blacks elsewhere?

We have amended the paragraph with additional text: “A review published in 2016 indicated that vitamin D status among different South African population groups and ethnicities was generally sufficient in children and adults, but was insufficient in many individuals over the age of 65 years [Norval et al 2016].”

When it is said that campaigns are “available on the Internet”, does that mean referred to or cited on the Internet, delivered to the public via the Internet, or both?

This is a good point for us to clarify. We have rephrased the sentence as follows: “In Southern Africa, information about skin cancer, its prevention and local campaigns to raise awareness about skin cancer that are available on the Internet (and therefore retrievable) are most common in South Africa and Namibia”

Perhaps text message interventions could be suggested as an option since it seems that more people have cell phones than Internet access, and texting interventions have shown promise elsewhere.

We agree and it is true that text message options are commonly used in Africa, including for reminders for antiretroviral drugs collection etc.

Climate

 “Projections for Southern Africa suggest that the number of hot days (≥27°C) will range between 81 and 296 per year, relative to 1961-1990, by 2100 [45].” This sentence is unclear unless it is meant that the number of hot days will increase, rather than range, since no comparison is provided.

We have amended this sentence as follows: “Projections for Southern Africa suggest that the number of hot days (≥27°C) will increase.”

“The estimates range from a modest 5.5% increase in SCC and a 2.9% increase in BCC per °C increase [46] to a 10% increase for every 2°C increase in ambient temperature [47].” Please revise this sentence to clarify it. I was initially confused because 5.5x2~10, but then I realized the 10% was probably referring to BCC only.

We have updated this information with a more recent study, continuing from Prof van der Leun’s work as follows: “There is some evidence that skin cancer rates may increase with a rise in ambient temperature, although the reasons why are not fully understood [53]. The latest estimates range from a modest 5.8% increase in SCC and a 2.8% increase in BCC according to ambient temperature increases projected between 2018 and 2100 [53] [54].”

What is the prevalence of the use of shipping containers and/or air conditioning in schools?

Unfortunately, this information is not quantitatively available. Anecdotal evidence and observations suggest shipping containers are being used more often than in the past. No information about how many schools have air conditioners is available; moreover, with the current electricity crisis in the country (i.e. load-shedding or rolling black-outs), it is likely that the few schools with air conditioning would not be able to use them regularly.

“…evidence suggests that the time-lapse between changes in ambient temperature and skin cancer incidence is about 60 years [51].” The paper cited here is quite old.

A recent reference is now added.

Sun protection

Some recommendations specify 30+ for sunscreen SPF.

We agree that some recommendations are for higher SPFs but, here we have relied on the WHO website https://www.who.int/uv/sun_protection/en/ as the source of up-to-date information.

What is the prevalence of sunlamp use in Southern Africa?

Very little research has been done to quantify sunbed prevalence in South Africa. One study (Wright CY, Albers PN, Reeder AI and Mathee A. Sunbeds and skin cancer risk: Quantifying a baseline estimate of sunbed facilities in South Africa prior to implementation of sunbed regulations. Pan African Medical Journal 2017;26:188. DOI: 10.11604/pamj.2017.26.188.10176) used the now ‘outdated’ Yellow Pages to look for adverts for sunbed parlours as well as a website recommending spas and salons that had sunbeds. Using such a weak methodology does not provide reliable estimates of sunbed prevalence in the country and hence these data remain unknown.

“Sun protection is critical for people with OCA.” Suggest adding “and other individuals with fair skin”.

Thank you for this suggestion which we have implemented.

Are there any data on outdoor workers in Southern Africa?

We have added the following sentence: A few studies have assessed outdoor worker sun exposure patterns in South Africa [Nkogatse MM, Ramotsehoa MC, Eloff FC and Wright CY. Solar ultraviolet radiation exposure and sun protection behaviours and knowledge among a high-risk and overlooked group of outdoor workers in South Africa. Photochemistry and Photobiology 2019; 95: 439-445; Makgabutlane, T and Wright CY. 2015. Real-time measurement of outdoor worker’s exposure to solar ultraviolet radiation in Pretoria. South African Journal of Science. 111(5/6):1-7. DOI: http://dx.doi.org/10.17159/sajs.2015/20140133.], and work is underway to illustrate sun protection use.

Caution about the use of the UVI as a skin cancer prevention tool for the public is warranted. See Prev Med.2019 Jun;123:71-83. doi: 10.1016/j.ypmed.2019.03.004. Epub 2019 Mar 4. Awareness, understanding, use, and impact of the UV index: A systematic review of over two decades of international research. Heckman CJ1, Liang K2, Riley M3.

We agree and we have amended the text in the manuscript in line with the WHO recommendations that the UVI is a resource to help plan outdoor activities to prevent overexposure to the Sun.

It seems that HIV+ individuals would be a potential target for sun protection efforts. Is it worth mentioning other skin cancers common among them such as kaposi’s sarcoma?

We have inserted a short paragraph to summarise the situation regarding Kaposi’s sarcoma as follows: “Another common skin tumour in HIV-positive individuals is Kaposi’s sarcoma although its incidence has declined markedly since the advent of antiretroviral therapy. There is limited evidence from epidemiological studies to indicate that exposure to solar UVR increases the risk of Kaposi’s sarcoma [Cahoon EK et al J Natl Cancer Inst 2017].”

Reviewer 2 Report

Dear Editor and Authors,

the paper “The Epidemiology of Skin Cancer and Public Health Measures for its Prevention in Southern Africa” presents an interesting overview of skin cancers epidemiology in a region for which currently not many data on these diseases are available in scientific literature.

Nevertheless, there are some sections of the manuscript that are not fully clear for the readers, especially considering the methodology by with the reported data are derived. I refer in particular to the section 6 “Mortality due to skin cancer in Southern Africa”, where it is not clear to me how these mortality data are collected, and it seems not coherent to report together CM and KCs mortalities, as, to my knowledge, mortality risk for BCC is approximately equal to 0, while a low mortality rate can be associated to SCC. On the other hand, mortality rates for CM are very relevant. This section needs more detailed information, as it can’t be considerate adequately informative at the current status.

Other minor suggestions as following:

 Table 2: I think that some information is not necessary considering the nature of the diseases presented, and in particular I don’t see the importance of reporting the mean night temperature in summer and winter. I suggest to delete this information.

Line 9-10 page 3: I think that a mention to the characteristics of the ozone layer in southern Africa should be important.

Lines 39-41 page 5: please add an adequate reference, as at my knowledge this sentence is not correct. In my experience BCC is more associated to intense intermittent exposure and also to cumulative exposure while CM is related to high intermittent exposure and repeated sunburns particularly in childhood, and only Lentigo Maligna is thought to be associated to cumulative UV exp (see Armstrong 2017).

Section 7: among the risk factors discussion, an important point is missed in particular for BCC and SCC etiology, that are highly associated to occupational solar UV exposure in outdoor workers (please see eg. Paulo et al. Env Int 2019; Modenese et al IJERPH 2018; John et al. JEADV 2016).

Author Response

Reviewer Two

Dear Editor and Authors,

the paper “The Epidemiology of Skin Cancer and Public Health Measures for its Prevention in Southern Africa” presents an interesting overview of skin cancers epidemiology in a region for which currently not many data on these diseases are available in scientific literature.

Nevertheless, there are some sections of the manuscript that are not fully clear for the readers, especially considering the methodology by with the reported data are derived. I refer in particular to the section 6 “Mortality due to skin cancer in Southern Africa”, where it is not clear to me how these mortality data are collected, and it seems not coherent to report together CM and KCs mortalities, as, to my knowledge, mortality risk for BCC is approximately equal to 0, while a low mortality rate can be associated to SCC. On the other hand, mortality rates for CM are very relevant. This section needs more detailed information, as it can’t be considerate adequately informative at the current status.

A new sentence has been added to address the first part of this comment. Regarding the mortality rates for skin cancer, the method by which the data in Table 4 were complied is outlined in the reference [6]. Although, as indicated in the text, the basis of the figures for each country is not clear, these represent the only results available to date.

South Africa has the most comprehensive cancer registry in the region, and, even here, we found it very difficult to obtain reliable and comprehensive data regarding CM mortality. The best we could do is sunmmarised in reference [21]. An outline of our findings is provided in the text.

Other minor suggestions as following:

 Table 2: I think that some information is not necessary considering the nature of the diseases presented, and in particular I don’t see the importance of reporting the mean night temperature in summer and winter. I suggest to delete this information.

We have deleted this information from the table.

Line 9-10 page 3: I think that a mention to the characteristics of the ozone layer in southern Africa should be important.

We have amended Line 9-10 page 3 “Solar UVR levels across Southern Africa depend on factors such as latitude, altitude, ozone, aerosols (e.g. sea salt spray, air pollution), albedo (reflection from snow or water) and cloud cover.” To highlight the importance of the ozone layer we have added the following to Line 15-17 page 3 “Over Southern Africa, total column ozone is at a maximum during the austral spring and decreases over the summer months when UVR is at a maximum. More southerly areas experience a higher variability of ozone throughout the year.”

Lines 39-41 page 5: please add an adequate reference, as at my knowledge this sentence is not correct. In my experience BCC is more associated to intense intermittent exposure and also to cumulative exposure while CM is related to high intermittent exposure and repeated sunburns particularly in childhood, and only Lentigo Maligna is thought to be associated to cumulative UV exp (see Armstrong 2017).

We have corrected the sentence relating to BCC and added more information about sun exposure and the risk of CM.

Section 7: among the risk factors discussion, an important point is missed in particular for BCC and SCC etiology, that are highly associated to occupational solar UV exposure in outdoor workers (please see e.g. Paulo et al. Env Int 2019; Modenese et al IJERPH 2018; John et al. JEADV 2016).

This point is now added near the beginning of the second paragraph of section 7 together with a new reference.

Round 2

Reviewer 1 Report

The authors were responsive to prior reviews. A couple of minor suggestions remain.

Suggest adding something about that anecdotally, it seems that use of shipping containers and/or lack of air conditioning may be common and/or increasing. And that rates of sunbed use in Southern Africa are unknown. 

I don’t see where this was added - We have added the following sentence: A few studies have assessed outdoor worker sun exposure patterns in South Africa [Nkogatse MM, Ramotsehoa MC, Eloff FC and Wright CY. Solar ultraviolet radiation exposure and sun protection behaviours and knowledge among a high-risk and overlooked group of outdoor workers in South Africa. Photochemistry and Photobiology 2019; 95: 439-445; Makgabutlane, T and Wright CY. 2015. Real-time measurement of outdoor worker’s exposure to solar ultraviolet radiation in Pretoria. South African Journal of Science. 111(5/6):1-7. DOI: http://dx.doi.org/10.17159/sajs.2015/20140133.], and work is underway to illustrate sun protection use.

Author Response

We thank the reviewer for helping us to see what we missed when updating the manuscript itself.

We have now included the text about the prevalence of shipping containers.

We did mention the limited studies on occupational sun protection with the sentence and references as follows: 

Patterns of sun exposure in outdoors workers have been assessed in limited studies [69, 70].

We hope this meets your suggestions.

Reviewer 2 Report

Dear Authors,

I appreciated your effort in improving the manuscript.

Nevertheless, one the sentence you added in the revision stage is "A distinction between death due to KC or CM was not made although it is likely that the majority was due to CM with a minority due to SCC, and none to BCC", and my opinion is that this sentence does not sound as an adequately scientific assumption.

On the other hand, you provided a very relevant reference on the trends in melanoma mortality in South Africa [21].

Based on the data from this paper, you can certainly do something more regarding a possible estimate of the percentage of melanoma deaths among all skin cancers deaths, in particular in South Africa, but for extrapolation also in the other Countries: please try to write a(few) more informative scientific sentence(s) discussing the numbers of melanoma deaths in South Africa and their burden, being more precise: you have all the data to do this.

Furthermore, you added a sentence at the end of the Discussion (page 9) stating that "Patterns of sun exposure in outdoors workers have been assessed in limited studies [68, 69]". This is not fully correct, maybe you can add "in Africa" after "studies". Then, you should mention that in many other studies e.g. from Europe or North America or Oceania relations between pattern of sun exposure and risk of adverse health effects have been considered: please see e.g. Modenese et al. J Eur Acad Dermatol Venerol 2016 Apr;30 Suppl 3:21-6 for Europe, Gies et al. Am J Ind Med. 2009 Aug; 52(8): 645–653. for North America, and Hammond et al. Public health 123(2):182-7 , 2009 for Oceania.

Best regards, the Reviewer

Author Response

Thank you for providing us with additional comments. For the first point, we had access to melanoma mortality data at a time when Statistics South Africa would release such data. We were never given data for SCC or BCC so it is difficult to calculate exact percentage of melanoma deaths among SCC+BCC deaths accurately. However, using the Stats SA latest death report, we found that of the 456 612 deaths in South Africa in 2016, 826 were from melanoma and 485 were from ‘other disorders of skin and subcutaneous tissue’ which includes SCC and BCC. Hence there were about twice (58%) as many CM deaths than KCs deaths in 2016. We have included this information in the manuscript. The citation is Stats SA, 2016, Mortality and causes of death in South Africa, 2016: Findings from death notification.

Thank you for the suggestion to mention the limitation of occupational assessments being in South Africa. We have added this text as well as your suggestion to mention that other studies around the world do exist, and we have included your recommended citations.